# The Multifaceted Role of Mitochondria in Angiogenesis

**DOI:** 10.3390/ijms26167960

**Published:** 2025-08-18

**Authors:** Sara Cannito, Ida Giardino, Maria d’Apolito, Massimo Pettoello-Mantovani, Francesca Scaltrito, Domenica Mangieri, Annamaria Piscazzi

**Affiliations:** 1Laboratory Medicine, Department of Clinical and Experimental Medicine, University of Foggia, 71122 Foggia, Italy; sara.cannito@unifg.it (S.C.); ida.giardino@unifg.it (I.G.); 2Medical Genetics, Department of Clinical and Experimental Medicine, University of Foggia, 71122 Foggia, Italy; maria.dapolito@unifg.it; 3Italian Academy of Pediatrics, 20126 Milan, Italy; massimo.pettoellomantovani@unifg.it; 4Department of Pediatrics, Institute for Scientific Research «Casa Sollievo», University of Foggia, 71122 Foggia, Italy; 5European Pediatric Association, Union of National European Pediatric Societies and Associations, 10115 Berlin, Germany; 6Residency Course in Pediatrics, Department of Medical and Surgical Sciences, University of Foggia, 71122 Foggia, Italy; francesca.scaltrito@unifg.it; 7Applied Biology, Department of Clinical and Experimental Medicine, University of Foggia, 71122 Foggia, Italy; domenica.mangieri@unifg.it

**Keywords:** mitochondria, angiogenesis, mitochondrial calcium transport, mtROS-HIF1a-VEGF axis, VEGFR2

## Abstract

Angiogenesis, the formation of new blood vessels from pre-existing ones, is crucial for various physiological and pathological conditions, including embryonic development, wound healing, tissue regeneration and tumor progression. While traditionally attributed to the actions of growth factors and their receptors, emerging evidence highlights the crucial regulatory roles of mitochondria in angiogenesis. This narrative review explores the multifaceted functions of mitochondria in endothelial cells, which are central to blood vessel formation. Beyond their classical role in ATP production, mitochondria contribute to angiogenesis through redox signaling, calcium homeostasis, biosynthetic activity, and reactive oxygen species (ROS) generation. These organelles help regulate key endothelial behaviors such as proliferation, migration, and tube formation through mechanisms that include mitochondrial calcium signaling and ROS-mediated stabilization of hypoxia-inducible factor-1α (HIF-1α), leading to increased vascular endothelial growth factor (VEGF) expression. Additionally, mitochondrial dynamics, dysfunction, and genetic factors are discussed for their influence on angiogenic outcomes. Understanding these complex mitochondrial functions opens new therapeutic avenues for modulating angiogenesis in diseases such as cancer and cardiovascular disorders.

## 1. Introduction

Angiogenesis, the formation of new blood vessels from pre-existing vasculature, is essential for various physiological processes, including embryonic development, wound healing, and tissue regeneration [1]. However, its dysregulation is a hallmark of numerous pathological conditions, such as tumor progression and cardiovascular diseases, highlighting the need for tightly regulated mechanisms to maintain tissue homeostasis [1]. Pathological angiogenesis is characterized by uncontrolled or aberrant blood vessel growth (Figure 1). This dysregulation commonly arises from an imbalance between pro-angiogenic (vessel-promoting) and anti-angiogenic (vessel-inhibiting) factors (Figure 2) [2].

Historically, angiogenesis research has primarily focused on the roles of growth factors and their receptors in promoting endothelial cell proliferation and migration. Recent advances, however, have led to a paradigm shift, revealing that mitochondria, traditionally regarded as the primary sites of ATP production via oxidative phosphorylation, play broader roles in regulating cellular processes relevant to angiogenesis [3,4,5,6]. Mitochondria are now recognized as critical signaling hubs within endothelial cells, the principal cellular component of blood vessels. They participate in redox signaling, calcium homeostasis, apoptosis, and inflammatory responses [3,4,5,6]. Accordingly, there is increasing recognition of their role as one of the actors in the regulation of the angiogenesis process [3,4,5]. This narrative review aims to provide a comprehensive overview of the multifaceted roles of mitochondria in angiogenesis. It will explore their signaling functions, their metabolic interactions within endothelial cells during vessel formation, the consequences of mitochondrial dysfunction and genetic alterations on angiogenic processes, and the therapeutic potential of targeting mitochondrial pathways to modulate blood vessel growth.

## 2. Metabolic Interplay in Angiogenesis: Balancing Glycolysis and Oxidative Phosphorylation

Endothelial cells, which form the inner lining of blood vessels, exhibit a unique metabolic profile characterized by a preferential reliance on glycolysis for adenosine triphosphate (ATP) production, even under normoxic conditions. This phenomenon is often referred to as the Warburg effect in the context of cancer cells, but its application to endothelial cells in angiogenesis highlights a similar metabolic reprogramming [1,4,5]. This seemingly counterintuitive metabolic choice offers several advantages to endothelial cells, particularly during the dynamic process of angiogenesis:Rapid ATP Production. Glycolysis, as an anaerobic process, can generate ATP at a faster rate compared with oxidative phosphorylation. This rapid energy production is crucial for the bursts of energy required for swift cell migration and proliferation, particularly at the leading edge of newly forming vessels [7].Biomass Synthesis. Glycolytic intermediates are shunted into anabolic pathways (e.g., pentose phosphate pathway, serine synthesis pathway). These pathways produce essential building blocks such as nucleotides, lipids, and amino acids, which are vital for rapid cell division and the extensive new vessel formation that characterizes angiogenesis [8].Adaptation to Hypoxia. This metabolic preference allows endothelial cells to adapt to hypoxic environments, which are frequently encountered during angiogenesis (e.g., in rapidly growing tissues or ischemic areas). By limiting their own oxygen consumption through reliance on glycolysis, endothelial cells may indirectly facilitate a more efficient transfer of available oxygen to the surrounding metabolically active tissues [7].

This metabolic shift towards glycolysis appears to be crucial for supporting the increased energy demands associated with the rapid cell proliferation and migration that characterize the formation of new blood vessels, particularly at the tip cells that lead the angiogenic sprout [9]. Key glycolytic enzymes, such as 6-phosphofructo-2-kinase/fructose-2, 6-bisphosphatase 3 (PFKFB3), have been shown to play a significant role in regulating the glycolytic flux and, consequently, angiogenesis [10].

Despite glycolysis serving as the dominant ATP-generating pathway in endothelial cells, mitochondria metabolism still plays a significant role during specific phases of angiogenesis or under certain physiological or pathological conditions [3].

Increasing evidence has suggested the importance of mitochondrial biology including metabolism, quality control, location, signaling regulation, and homeostasis in controlling endothelial cell (EC) permeability, tone, migration, and proliferation under both physiological and pathophysiological conditions [11,12,13].

In pathological conditions such as cancer, the metabolic adaptations in endothelial cells can be significantly perturbed, contributing to the abnormal and often excessive angiogenesis that fuels tumor growth [14]. The “angiogenic switch” in tumors is often accompanied by increased glucose uptake and glycolysis to support the rapid proliferation of both tumor cells and the endothelial cells that form the tumor vasculature (Figure 2) [15]. Mitochondrial metabolism in these pathological settings can be altered in complex ways, with some evidence suggesting that mitochondrial dysfunction might even enhance angiogenic potential in certain cancer types [16]. The precise role of mitochondrial metabolism in pathological angiogenesis appears to be highly context-dependent, varying depending on the specific disease and cellular environment.

## 3. Mitochondrial Functions in Endothelial Cells: More than Just Powerhouses

### 3.1. Mitochondria as Biosynthetic Hubs

During angiogenesis, the formation of new blood vessels requires rapid cell division. Beyond their role in energy production, mitochondria act as metabolic hubs for endothelial cell (EC) proliferation during angiogenesis [3]. They are involved in the synthesis of various essential molecules, including amino acids, lipids, and heme, which are crucial building blocks for cell growth, proliferation, and other processes integral to angiogenesis [3,4]. Fatty acid oxidation (FAO), controlled by CPT1A (Carnitine Palmitoyltransferase 1 A), provides acetyl-CoA for the TCA cycle and, importantly, generates reducing equivalents (NADH and FADH2) that fuel oxidative phosphorylation (OXPHOS). This indirectly supports de novo nucleotide synthesis required for proliferation and vessel branching. Quiescent ECs tend to utilize FAO to maintain redox homeostasis, while angiogenic ECs utilize FAO to support nucleotide synthesis for sprouting [17]. This indicates a dynamic regulation of FAO based on the EC’s functional state. Notably, CPT1A deficiency impairs EC proliferation and angiogenesis in vivo without affecting migration or tip cell numbers. Uniquely for ECs, FAO contributes carbon to non-lipid biomass, highlighting its diverse metabolic contributions.

ECs also exhibit high glutamine consumption. Glutamine is an important alternative carbon and nitrogen source for ECs, primarily metabolized within mitochondria through glutaminolysis. Glutaminase 1 (GLS1) is a mitochondrial enzyme that converts glutamine to glutamate, initiating glutaminolysis. Glutamine metabolism significantly contributes to the TCA cycle via anaplerosis (the replenishment of TCA cycle intermediates), thereby supporting EC proliferation, especially by providing carbons for biomass synthesis [3]. Glutamine depletion or GLS1 inhibition severely impairs EC proliferation and macromolecular biosynthesis. Glutamine also contributes to asparagine synthesis via ASNS (Asparagine Synthetase), which is crucial for growth, mTORC activation, protein synthesis, and reducing endoplasmic reticulum (ER) stress. Amino acid metabolism further impacts tip cell behavior; GLS1 inhibition reduces tip cell numbers and impairs their competitive positioning [18]. For instance, the final steps of heme biosynthesis occur within the mitochondria, where ferrochelatase (FECH) inserts Fe^2+^ (ferrous iron) into protoporphyrin IX to produce protoheme IX (commonly known as heme). This process has been shown to be linked to both mitochondrial function and angiogenesis [14].

### 3.2. Mitochondria Respiration and ROS Production

As previously reported, endothelial cells undergoing angiogenesis exhibit a high rate of glycolysis [1]. While a substantial proportion of pyruvate, the end-product of glycolysis, is shunted to lactate, a critical fraction nonetheless translocates into the mitochondria via the mitochondrial pyruvate carrier (MPC) to fuel the TCA cycle. This residual OXPHOS activity is indispensable for preserving mitochondrial membrane potential and ensuring overall mitochondrial health, both of which are prerequisites for endothelial cell (EC) activation and robust angiogenesis. Several studies have demonstrated that inhibiting mitochondrial respiration can impair endothelial cell proliferation and angiogenesis, highlighting the indispensable role of mitochondrial metabolism beyond just ATP production [3,4,5].

Reactive oxygen species (ROS) are a natural byproduct of oxidative phosphorylation, but under normal metabolic conditions, only about 0.1% of the total oxygen consumed in the mitochondria is converted to mitochondrial ROS (mtROS) [5]. The principal sources of mtROS are Complexes I and III of the mitochondrial electron transport chain (ETC). These complexes contribute to ROS generation during oxidative phosphorylation; however, their exact quantitative role in ECs remains uncertain due to the presence of multiple ROS-generating pathways [5]. One such prominent source is NADPH oxidase 4 (NOX4), which is highly expressed in ECs and has emerged as a key regulator of redox signaling within the mitochondria [19]. Of particular interest is the concept of ROS-induced ROS release, a feedforward mechanism in which ROS originating from non-mitochondrial sources (e.g., NOX enzymes) initiate mitochondrial ROS generation [20]. For example, Angiotensin II stimulates NOX-dependent ROS production that can activate mitochondrial channels, such as the ATP-sensitive potassium channel (mitoK_ATP), leading to mitochondrial membrane depolarization and the opening of the permeability transition pore. This process enhances mtROS production and, in turn, modulates endothelial nitric oxide levels, possibly serving as a protective adaptation against nitrosative stress [21].

Specific mitochondrial proteins, such as p66Shc and UQCRB, have been implicated in the generation and signaling of ROS in the context of angiogenesis (Table 1).

The growth factor adaptor protein p66Shc has been shown to oxidize cytochrome c to generate hydrogen peroxide. Upon activation by stress signals such as hyperglycemia or pro-apoptotic stimuli, p66Shc translocates to the mitochondrial intermembrane space. There, it triggers ROS-dependent signaling cascades implicated in endothelial dysfunction and apoptosis. Functionally, p66Shc acts as a redox transducer, integrating environmental and metabolic cues into mtROS signals that influence vascular homeostasis (Table 1) [22].

Ubiquinol-cytochrome c reductase binding protein (UQCRB), a subunit of mitochondrial Complex III, is essential for electron transport and the generation of ROS within mitochondria [3,23]. The UQCRB subunit plays a role in mtROS production, potentially as a modulator of electron flux through Complex III, which can influence the lifetime of ubisemiquinone, thereby controlling the levels of mtROS produced (Table 1) [3].

Sirtuin 3 (SIRT3), primarily located in the mitochondrial matrix, is involved in regulating the production of ROS and aspects of mitochondrial metabolism [3]. Studies suggest that SIRT3 plays a pro-angiogenic role, as its overexpression promotes VEGF (Vascular Endothelial Growth Factor) production and angiogenesis, while its deficiency has the opposite effect (Table 1) [24].

In summary, ROS production by endothelial mitochondria can be modulated by different mechanisms, and it is known to play a significant role in regulating cellular signaling responses. While excessive ROS can lead to oxidative stress and endothelial dysfunction, at low levels, ROS act as important signaling molecules involved in various cellular processes, including angiogenesis [1,25,26,27,28].

**Table 1 ijms-26-07960-t001:** Key Mitochondrial Proteins Involved in Angiogenesis.

Protein Name	Location Within Mitochondria	Primary Function	Role in Angiogenesis	Ref.
**POLRMT** **(RNA Polymerase** **Mitochondrial)**	Mitochondrial matrix	Regulates mitochondrial transcription and oxidative phosphorylation	Anti-angiogenic: Silencing/knockout Impedes proliferation, migration, and tube formation. Linked to pathological angiogenesis in diabetic retinopathy.	[29,30]
**TIMM44** **(Translocase of Inner Mitochondrial Membrane 44)**	Inner mitochondrial membrane	Essential for mitochondrial integrity and function	Anti-angiogenic: Silencing/blocking Inhibits proliferation, migration, and tube formation in vitro and in vivo. A potential therapeutic target for abnormal Angiogenesis.	[31]
**SIRT3 (Sirtuin 3)**	Outer mitochondrial membrane	Regulates ROS formation and glycolysis	Anti-proliferative: Knockdown decreases ATP production and inhibits mTOR activity, affecting cell proliferation.	[24]
**VDAC1** **(Voltage-Dependent Anion Channel 1)**	Outer mitochondrial membrane	Regulates metabolite exchange (ATP/ADP)	Anti-proliferative: Knockdown decreases ATP production and inhibits mTOR activity, affecting cell proliferation.	[32,33]
**Drp1** **(Dynamin-** **Related Protein 1)**	Outer mitochondrial membrane	Mediates mitochondrial fission	Anti-migratory/proliferative: Knockdown impairs cell migration and proliferation. Inhibition can protect against ischemia–reperfusion injury.	[34]
**p66Shc (66 kDa proto-oncogene Src homologous-collagen homolog adaptor protein)**	Inner mitochondrial membrane	Involved in ROS dependent signaling	Pro-angiogenic: Critical role in ROS-dependent VEGF signaling and angiogenesis. Regulates oxidative stress and apoptosis pathways.	[22]
**UQCRB** **(Ubiquinol-Cytochrome c Reductase Binding Protein)**	Inner mitochondrial membrane (Complex III subunit)	Regulates electron transport and ROS Production	Anti-angiogenic: Inhibition reduces VEGF-mediated cell proliferation and angiogenesis.	[23]
**ALDH2 (Aldehyde** **Dehydrogenase 2)**	Mitochondrial matrix	Antioxidant by detoxifying aldehydes	Pro-angiogenic: Overexpression promotes endothelial cell migration, proliferation, and angiogenesis.	[35]
**CypD (Cyclophilin D)**	Mitochondrial matrix	Regulates calcium levels, energy metabolism, and apoptosis	Pro-angiogenic: Deficiency can increase VEGF-induced proliferation and angiogenesis.	[33,36]

## 4. Mitochondrial Signaling and Angiogenesis: Orchestrating New Vessel Growth

### 4.1. Mitochondria as Central Oxygen Sensors in Endothelial Cells

In endothelial cells, mitochondria are pivotal oxygen sensors within the vascular system [6]. They detect fluctuations in oxygen availability and initiate signaling cascades that profoundly influence angiogenesis [5]. During hypoxic conditions, such as those arising from tissue injury or tumor development, there is a notable increase in the production of mitochondrial reactive oxygen species (mtROS). Specifically, Complex III of the mitochondrial electron transport chain is central to this oxygen-sensing mechanism. Under hypoxia, the elevated generation of mtROS at Complex III activates downstream pathways that promote angiogenesis [22,23,24,37]. These mtROS molecules serve as critical signaling agents by stabilizing Hypoxia-Inducible Factor-1 alpha (HIF-1α). They achieve this by inhibiting prolyl hydroxylase enzymes, which normally hydroxylate HIF-1α under normoxic conditions (normal oxygen levels), thereby marking it for degradation via the proteasome [5]. Once stabilized, HIF-1α translocates to the nucleus, where it functions as a transcription factor, regulating gene expression. In the nucleus, HIF-1α upregulates the expression of numerous genes involved in the cellular hypoxic response, most notably the gene encoding the Vascular Endothelial Growth Factor (VEGF) [5]. The VEGF, a potent pro-angiogenic growth factor (a molecule that promotes the formation of new blood vessels), is subsequently secreted by cells and binds to its primary receptor, VEGF Receptor 2 (VEGFR2), (Figure 3) located on endothelial cells [1]. Activation of VEGFR2 triggers key downstream signaling cascades, including the extracellular signal-regulated kinase (ERK) and protein kinase B (Akt) pathways. These pathways collectively promote endothelial cell proliferation, migration, and survival, ultimately culminating in the formation of new blood vessels [37].

### 4.2. Mitochondria as Responsible of Intracellular Calcium Homeostasis

In addition to the well-established mtROS–HIF-1α–VEGF axis, mitochondrial calcium (Ca^2+^) signaling plays a crucial role in regulating endothelial cell (EC) behavior during angiogenesis [6]. Fluctuations in intracellular calcium levels, largely governed by mitochondrial uptake and release, significantly influence EC proliferation, migration, and tube formation, all of which are vital for new blood vessel development [32]. Mitochondria are central to maintaining intracellular calcium homeostasis in endothelial cells [3]. It is estimated that approximately 75% of the total intracellular Ca^2+^ pool is stored in the endoplasmic reticulum (ER), while the remaining 25% resides in mitochondria. Despite these distinct calcium reservoirs, a high degree of functional cooperation exists between the ER and mitochondria in regulating intracellular Ca^2+^ flux [33]. In many cell types, mitochondria transiently sequester cytosolic Ca^2+^, which is later recaptured by the ER, highlighting a dynamic interplay between these organelles in calcium signaling and homeostasis. Specific mitochondrial proteins mediate calcium (Ca^2+^) transport across mitochondrial membranes, thereby modulating intracellular calcium signaling pathways that are critical for angiogenesis [32]. Mitochondrial Ca^2+^ uptake is primarily governed by the mitochondrial calcium uniporter complex (MCUC), which comprises the pore-forming mitochondrial calcium uniporter (MCU) and a set of regulatory subunits including MCUb, MICUs, EMRE, and MCUR1 [32]. Notably, MCU also functions as a redox sensor; under hypoxic conditions, S-glutathionylation of MCU enhances Ca^2+^ uptake, leading to increased mitochondrial reactive oxygen species (mtROS) production and heightened cellular sensitivity to death signals. Mitochondrial Ca^2+^ efflux occurs through specific transporters such as the mitochondrial Na^+^/Li^+^/Ca^2+^ exchanger (NCLX) and the H^+^/Ca^2+^ exchanger (HCX), as well as through the mitochondrial permeability transition pore (mPTP). The mPTP is a multi-protein complex composed of voltage-dependent anion channel 1 (VDAC1), a translocator protein (TSPO), and cyclophilin D (CypD) (Figure 4) [32,33].

Mitochondria also maintain close physical and functional contact with the endoplasmic reticulum (ER) via specialized regions known as mitochondria-associated membranes (MAMs) [36,38]. These microdomains facilitate efficient Ca^2+^ transfer from the ER to mitochondria, which is essential for sustaining mitochondrial bioenergetics under physiological conditions. The inositol 1,4,5-trisphosphate receptor (IP_3_R), embedded in the ER membrane, serves as the principal channel for ER Ca^2+^ release. In endothelial cells, mitochondria can directly associate with IP_3_Rs at MAMs to enable highly efficient Ca^2+^ transfer [39,40]. This Ca^2+^ signaling axis plays a pivotal role in regulating mitochondrial oxidative metabolism and angiogenic processes.

### 4.3. Mitochondrial Dynamics in Angiogenesis

Mitochondria are highly dynamic organelles that continuously change their morphology, distribution, and functional state through the coordinated processes of fusion and fission [34]. These structural alterations are not merely passive events but are crucial for maintaining mitochondrial health, cellular bioenergetics, and signaling. Fusion, which promotes the interconnection of mitochondria, is mediated by mitofusin 1 and 2 (Mfn1/2) on the outer mitochondrial membrane and optic atrophy 1 (Opa1) on the inner mitochondrial membrane. Fission, the division of mitochondria, is driven primarily by dynamin-related protein 1 (Drp1), which is recruited to the mitochondrial surface by adaptors such as Fis1, Mff, and MiD49/51 [34,41]. Recent studies unveiled the pivotal role of mitochondrial dynamics in regulating both physiological angiogenesis during development and pathological angiogenesis in diseases such as cancer [42,43]. The processes of fission and fusion are tightly linked to endothelial cell (EC) function through their effects on redox signaling, energy distribution, and metabolic adaptation [44]. In sprouting angiogenesis, ECs show an increase in Drp1-dependent mitochondrial fission, leading to the elevated production of mitochondrial reactive oxygen species (mtROS) [45,46]. These ROS act as signaling molecules that stabilize hypoxia-inducible factor 1-alpha (HIF-1α) and promote the expression of the vascular endothelial growth factor (VEGF), a key pro-angiogenic factor [41,45]. In the tumor microenvironment, mitochondrial dynamics are frequently dysregulated to favor pathological angiogenesis. Cancer cells often display hyperactivation of Drp1, resulting in excessive mitochondrial fragmentation and augmented VEGF secretion, which in turn promotes the angiogenic potential of surrounding stromal and endothelial cells. [43]. Conversely, promoting mitochondrial fusion, such as through increased Opa1 expression, dampens mtROS signaling and reduces angiogenic activity [41]. In this context, Herkenne et al. [47] provided compelling evidence that Opa1 is not only essential for maintaining mitochondrial fusion but also plays a direct and non-redundant role in angiogenesis. Using genetic models, they demonstrated that the loss of Opa1 in endothelial cells leads to impaired mitochondrial cristae remodeling, defective calcium uptake, and reduced EC proliferation and migration, ultimately resulting in defective blood vessel formation during development and in tumors. Importantly, Opa1-deficient ECs failed to activate necessary angiogenic transcriptional programs, independently of ATP production, suggesting a signaling-specific function of mitochondrial morphology in vascular growth. Mitochondrial trafficking is another essential component of EC angiogenic behavior. Mitochondria are actively repositioned toward the leading edge of migrating ECs, where localized ATP generation supports cytoskeletal remodeling and directional cell movement. This dynamic redistribution facilitates tip cell specification and efficient vascular branching [44].

### 4.4. Mitochondrial Quality Control

Additionally, mitochondrial dynamics are interconnected with quality control processes such as mitophagy and mitochondrial biogenesis, forming an integrated regulatory axis that ensures mitochondrial integrity and function in ECs under stress. The orchestration of these mechanisms ensures proper vascular adaptation, particularly under hypoxic or inflammatory conditions. Mitophagy is a selective form of autophagy that targets damaged or dysfunctional mitochondria for degradation [48]. This quality control process is mediated by proteins such as PTEN-induced kinase 1 (PINK1) and Parkin, which accumulate on depolarized mitochondria and label them for sequestration within autophagosomes [48]. Mitophagy plays a critical role in maintaining mitochondrial quality by preventing the accumulation of dysfunctional mitochondria that generate excessive ROS and compromise cellular metabolism and signaling. Dysregulation of mitophagy can lead to elevated oxidative stress and impaired endothelial function, ultimately affecting angiogenesis [49].

Given the integral role of mitochondrial dynamics and quality control in vascular homeostasis, the therapeutic modulation of mitophagy and related mitochondrial pathways present a promising strategy for controlling angiogenesis in various pathological conditions, including cancer, ischemic disease, and chronic inflammation (Table 2) [5].

### 4.5. Extracellular Vesicles (EVs): Mitochondrial Modulators in Vascular Health

Extracellular vesicles (EVs) have emerged as powerful modulators of mitochondrial activity, especially in endothelial cells (ECs), with significant implications for angiogenesis and vascular repair. EVs are nanoscale, membrane-bound particles, typically ranging from 30 to 120 nm in diameter, which play an essential role in intercellular communication. They transport bioactive cargo such as proteins, growth factors, microRNAs, and nucleic acids, delivering them to recipient cells and influencing a variety of signaling pathways [50].

Exosomes, a well-studied subtype of EVs, have been shown to influence mitochondrial function under both normal and pathological conditions [51]. For instance, adipose-derived EVs enhanced mitochondrial respiration and ATP production in human umbilical vein endothelial cells (HUVECs), outperforming bone marrow-derived EVs in promoting angiogenesis and tube formation [52]. This enhancement of mitochondrial function is partially attributed to the activation of the SIRT3/MnSOD pathway, as shown in exosomes from adipose-derived mesenchymal stem cells [53].

Further supporting their role in mitochondrial regulation, exosomes containing miR-210—released from endothelial progenitor cells (EPCs)—attenuated mitochondrial dysfunction in ECs subjected to hypoxia/reoxygenation injury. These exosomes reduced oxidative stress and apoptosis while promoting angiogenesis, largely through the preservation of mitochondrial function [54].

Beyond influencing mitochondrial signaling, EVs can also transfer mitochondrial content directly. EVs have been reported to contain mitochondrial components including membrane proteins, enzymes, and even mitochondrial DNA capable of modulating the metabolic state of recipient cells. For example, EPC-derived EVs transferred mitochondrial material to brain endothelial cells following oxygen glucose deprivation, increasing the mitochondrial DNA copy number, ATP production, and angiogenic activity [55]. Interestingly, microvesicles, a larger class of EVs (100–1000 nm), appear especially potent in modulating mitochondrial function. Microvesicles derived from a human cerebral microvascular endothelial cell line (hCMEC/D3) significantly upregulated mitochondrial activity, as shown by an increased oxygen consumption rate (OCR) and extracellular acidification rate (ECAR) under hypoxic conditions. In contrast, smaller EVs had minimal effects, suggesting that microvesicles may be more effective mitochondrial carriers [56].

Together, these findings point to an exciting therapeutic potential: using EVs, especially those enriched in mitochondrial elements, to restore endothelial function and stimulate angiogenesis. The effectiveness of this strategy depends greatly on the source of the EVs and their molecular composition, opening the door to tailored regenerative therapies in cardiovascular and ischemic diseases.

## 5. The Impact of Mitochondrial DNA Mutation on Angiogenesis: When Power Fails

Mitochondrial DNA (mtDNA) mutations are increasingly recognized as modulators of angiogenesis, but their impact diverges dramatically depending on whether the context is ischemic or malignant. mtDNA is particularly vulnerable to oxidative damage due to its proximity to respiratory chain complexes and limited DNA repair capacity, making it a hotspot for pathogenic mutations with variable angiogenic outcomes [57,58,59,60]. These mutations disrupt core mitochondrial functions, including ATP production, ROS regulation, and mitochondrial-nuclear signaling, and their consequences vary based on the mutation class, tissue type, heteroplasmy level, and the availability of compensatory metabolic programs.

### 5.1. Mitochondrial DNA Mutations: A Context-Dependent Modulator of Angiogenesis in Ischemia vs. Cancer

In ischemic diseases such as coronary artery disease (CAD), peripheral artery disease, and critical limb ischemia, effective endothelial cell (EC) responses depend on robust mitochondrial bioenergetics and redox balance. Here, mtDNA mutations tend to have a detrimental effect on angiogenesis, impairing neovascularization by compromising oxidative phosphorylation (OXPHOS) and disrupting ROS-mediated pro-angiogenic signaling. For example, the pathogenic tRNA^Thr 15927G>A mutation associated with CAD impairs mitochondrial protein translation, reduces ATP generation, and suppresses VEGF-A/VEGFR2 signaling in ECs, resulting in reduced migration, tube formation, and vascular repair [57]. Similarly, mutations in nuclear genes affecting mitochondrial function, such as NFU1, a key Fe-S cluster assembly protein, lead to impaired pyruvate dehydrogenase (PDH) activity and Complex I–III dysfunction, culminating in EC mitochondrial failure and decreased capillary density [61]. In ischemic settings, these mutations compromise the angiogenic machinery, especially when heteroplasmy levels exceed functional thresholds. The limited metabolic plasticity of ischemic ECs means that glycolysis cannot fully compensate for mitochondrial impairment, leading to stalled or regressive vascular growth [62].

In stark contrast, tumor cells frequently tolerate, or even exploit, mtDNA mutations to their advantage. Many tumors display remarkable metabolic adaptability, allowing them to sustain or even enhance angiogenesis despite mitochondrial dysfunction. This flexibility is partly due to the Warburg effect, in which tumor cells favor aerobic glycolysis, as well as enhanced glutaminolysis, lipid scavenging, and mitochondrial crosstalk with stromal cells, including horizontal mtDNA transfer [63]. In ovarian cancer xenograft models, pathogenic mutations in mtDNA respiratory genes are correlated with increased glycolysis and a higher reliance on glucose metabolism, which renders tumors more sensitive to anti-VEGF therapy [64].

Likewise, in non-small cell lung cancer (NSCLC), frameshift mutations in ND6 (e.g., 13885insC) upregulate the VEGF, CXCL12, and MMP-9, promoting EC recruitment, extracellular matrix remodeling, and metastatic neovascularization [65]. In these contexts, mtDNA mutations can stabilize HIF-1α via altered redox signaling without completely disrupting respiration, thereby enhancing angiogenesis in hypoxic tumor microenvironments. Moderate heteroplasmy levels may be particularly effective in tuning redox-sensitive pathways while preserving sufficient mitochondrial output to sustain tumor viability (Table 3) [66,67].

### 5.2. Cell-Type Specificity: Endothelial vs. Tumor Cell Responses

The cellular context profoundly influences how mtDNA mutations affect angiogenesis. In ECs, impaired mitochondrial function typically reduces migration, proliferation, and angiogenesis [61]. Endothelial cells rely on tightly regulated mitochondrial dynamics and transcription to coordinate migration, tip-cell behavior, and vessel stabilization. In this setting, mtDNA defects reduce mitochondrial trafficking to the lamellipodia, impair cytoskeletal remodeling, and disrupt tip–stalk cell balance, hallmarks of failed angiogenesis [47].

Tumor cells, however, exploit mitochondria not just for energy, but as signaling hubs. ROS, NAD^+^/NADH ratios, and metabolites like succinate and fumarate function as oncometabolites, stabilizing HIF-1α and activating angiogenic pathways. mtDNA mutations in this context serve as pro-angiogenic stressors, not liabilities. Tumor cells can secrete pro-angiogenic factors or remodel the microenvironment to sustain vascular support even under mitochondrial stress [65]. Thus, angiogenic outcomes depend not only on the mutation itself but also on the cellular context and crosstalk between compartments in the tumor microenvironment.

### 5.3. Heteroplasmy Level: High vs. Low Mutation Burden

Moreover, heteroplasmy, the proportion of mutant mtDNA copies relative to the wild type, critically shapes phenotypic outcomes.

In ischemia, high heteroplasmy levels (>80%) in ECs often exceed the bioenergetic threshold, triggering apoptosis, senescence, or metabolic collapse. These outcomes are incompatible with the high energy demands of angiogenic sprouting.

In tumors, however, intermediate heteroplasmy (30–60%) may be optimal for generating adaptive stress. Partial OXPHOS disruption at these levels promotes ROS-dependent angiogenic signaling without fully compromising mitochondrial function [67]. Tumor cells may even select for heteroplasmic stability to sustain this advantage. In tumors, the heteroplasmy threshold modulates not just mitochondrial function but also therapeutic responses—tumors with high heteroplasmy often demonstrate increased sensitivity to oxidative and anti-angiogenic stressors, while low heteroplasmy may support metabolic flexibility and resistance.

### 5.4. Compensatory Mechanisms: Metabolic Reprogramming and mtDNA Transfer

The capacity for metabolic adaptation and mitochondrial rescue mechanisms significantly distinguishes endothelial cells (ECs) in ischemic conditions from those in the tumor microenvironment. These differences are critical in determining the extent to which cells can withstand mitochondrial dysfunction and continue to support angiogenic processes.

In ischemic tissues, ECs primarily rely on oxidative phosphorylation (OXPHOS) under normoxic conditions and exhibit relatively limited metabolic plasticity. While hypoxic stress can stimulate a shift toward glycolysis, this reprogramming is often insufficient to fully compensate for defects in mitochondrial respiration caused by mtDNA mutations or depletion. This vulnerability is compounded by the fact that ECs under ischemic stress rarely engage in horizontal mitochondrial DNA (mtDNA) transfer, a process increasingly recognized as a cellular rescue mechanism. The lack of such compensatory mtDNA exchange, via mechanisms like tunneling nanotubes, renders ischemic ECs more susceptible to sustained mitochondrial dysfunction, impaired energy production, and compromised angiogenic capacity [60].

In contrast, tumor cells display marked metabolic flexibility that enables them to circumvent mitochondrial dysfunction and maintain angiogenic signaling despite substantial mtDNA damage. These cells can switch to aerobic glycolysis providing ATP and biosynthetic precursors independent of OXPHOS. Additionally, tumors exploit alternative metabolic pathways such as glutaminolysis, which replenishes tricarboxylic acid (TCA) cycle intermediates, and fatty acid oxidation (β-oxidation), which supports mitochondrial function under stress. Notably, cancer cells have demonstrated the ability to acquire functional mtDNA from surrounding stromal or immune cells through horizontal transfer mechanisms including tunneling nanotubes. This exchange enables the partial restoration of mitochondrial respiration and reinforces tumor cell survival, proliferation, and angiogenesis under otherwise non-permissive metabolic conditions [68]. In addition, recent insights suggest that mtDNA damage and mutations contribute to mitochondrial quality control mechanisms that further influence angiogenic behavior. The interplay between mitophagy, mtDNA integrity, and angiogenesis has been explored in the context of both ischemia and cancer. For instance, activation of PGC-1α-driven mitochondrial biogenesis may partially restore function in ischemic tissue, while in tumors, elevated mitophagy can facilitate metabolic rewiring under hypoxic conditions [8]. Additionally, circulating mtDNA can serve as a damage-associated molecular pattern (DAMP), modulating inflammatory angiogenesis through TLR9 (Toll-like receptor 9) and cGAS-STING pathways [69]. Conversely, the absence of mtDNA exchange in ischemic ECs, as reported by Wang et al., suggests a critical vulnerability in non-cancerous cells. While this contrast underscores a key distinction between cancer and ischemic tissues, the mechanistic reasons for this disparity remain underexplored.

### 5.5. Integration of POLRMT Regulation: A Central Transcriptional Control Node

POLRMT, the mitochondrial RNA polymerase, governs the transcription of mtDNA encoded genes and is emerging as a key regulator of angiogenesis, capable of amplifying or compensating for mitochondrial dysfunction depending on its expression and activity level [70,71]. Moreover, dysregulated POLRMT expression has been implicated in pathological neovascularization, such as in diabetic retinopathy, underscoring its dual role in physiological and aberrant angiogenesis [29]. Depletion of POLRMT in endothelial cells reduces proliferation, migration, and tube formation, mimicking the effects of pathogenic mtDNA mutations [71]. In ischemia, POLRMT downregulation reduces mitochondrial transcripts and impairs EC respiration and vessel growth [71]. Conversely, POLRMT overexpression enhances mitochondrial biogenesis, oxidative phosphorylation, and angiogenic function, reinforcing its role as a transcriptional bottleneck linking mitochondrial health to vascular growth [65]. In cancer, POLRMT may be preserved or upregulated, sustaining the expression of the remaining functional mtDNA loci to support ROS signaling and angiogenesis [68].

In conclusion, mtDNA mutations represent a double-edged sword in angiogenesis, a pathological brake in ischemia but a context-dependent accelerator in tumors. Future research should dissect the mutation-specific thresholds that determine angiogenic outcomes, explore cell-type-specific responses in the vascular niche, and develop selective strategies that exploit tumor-specific vulnerabilities without impairing physiological angiogenesis. Longitudinal studies using lineage tracing, single-cell genomics, and spatial transcriptomics will be essential to resolve these context-dependent roles and enable targeted mitochondrial interventions that are safe and effective across disease spectrums.

## 6. Therapeutic Potential of Targeting Mitochondria for Angiogenesis Modulation

Given their central role in cellular metabolism and signaling, mitochondria represent an attractive therapeutic target for the modulation of angiogenesis across a spectrum of diseases [72]. In clinical contexts where angiogenesis is insufficient, such as wound healing, ischemic cardiovascular disease, or neurovascular degeneration, enhancing mitochondrial function may restore angiogenic capacity. Conversely, in disorders characterized by excessive or pathological neovascularization, such as cancer and diabetic retinopathy, strategically inhibiting mitochondrial activity may suppress abnormal blood vessel formation [35]. In mitochondria targeting medicine, a first layer of complexity is introduced by tumor heterogeneity and the co-evolution of metabolic phenotypes within the tumor microenvironment. Some tumor subclones with severe mtDNA defects may act as metabolic donors or scavengers, facilitating the survival and angiogenesis in neighboring cells. Furthermore, immune cells in the tumor milieu may respond differently to mtDNA-encoded DAMPs (damage-associated molecular patterns), thereby influencing angiogenesis via paracrine cytokine signaling [73]. This raises important questions about the systemic impact of targeting mitochondrial function in cancer, particularly the risk of inadvertently disrupting beneficial pro-angiogenic responses in ischemic tissues. Another layer of complexity arises from the need for precise delivery of therapeutic or diagnostic agents specifically to mitochondria. This strategy is essential for enhancing treatment efficacy, minimizing off-target effects, and overcoming challenges such as drug resistance, which frequently hinder conventional therapies. The core principle enabling mitochondrial targeting lies in the highly negative mitochondrial membrane potential (ΔΨm), which typically ranges from –140 to –180 mV. This strong electrochemical gradient acts as an electrophoretic driving force, attracting positively charged molecules and enabling their accumulation in the mitochondrial matrix by up to 1000-fold. This inherent selectivity is particularly advantageous in cancer, where tumor cells often exhibit an even higher ΔΨm, thereby allowing preferential mitochondrial targeting in malignant tissues [74]. To exploit this property, several mitochondrial-targeting moieties have been developed, among which triphenylphosphonium (TPP) is the most widely used. TPP is a lipophilic cation that readily traverses mitochondrial membranes in response to ΔΨm. It has been successfully conjugated to various functional agents, including anticancer drugs, antioxidants (e.g., TPP-HT), and sensor molecules, greatly enhancing their bioactivity and mitochondrial specificity [75]. A significant development in mitochondrial targeting medicine is the increasing use of nanoparticles (NPs). Nanoparticles offer promising avenues for delivering drug payloads and sensors to mitochondria, effectively overcoming challenges such as low solubility, poor bioavailability, and issues with poor biodistribution. This approach facilitates more selective, precise, and safer delivery, potentially with fewer side effects [76,77].

### 6.1. Mitochondria-Targeted Antioxidants

Mitochondria-targeted antioxidants represent a promising class of therapeutic agents. By modulating reactive oxygen species (ROS) levels within the mitochondria, these compounds can fine-tune angiogenic signaling and improve endothelial cell (EC) viability [6]. For example, N-acetylcysteine (NAC), a precursor of the antioxidant glutathione, has demonstrated vascular-protective effects and improved EC function in various models of vascular disease [78,79].

**MitoQ**, mitochinone mesilato, a modified version of Coenzyme Q10 (CoQ10) that, beyond its critical role in electron transport within mitochondria, exhibits potent antioxidant capabilities. MitoQ has been shown to improve endothelial function in older adults by significantly lowering mitochondrial ROS (mtROS). This beneficial effect is partly attributed to reduced circulating levels of oxidized low-density lipoprotein (oxLDL), which contributes to vascular oxidative stress and endothelial dysfunction, thus directly demonstrating a mechanism by which mitochondrial antioxidants can support vascular health and, by extension, angiogenic processes [80]. Moreover, in cardiometabolic diseases MitoQ has been shown to reduce mitochondrial oxidative stress, prevent impaired mitochondrial dynamics, and increase mitochondrial turnover. Crucially, MitoQ supplementation can improve the expression of the vascular endothelial growth factor (VEGF) and neurogenic factors in hippocampal tissue, providing a direct link to improved neurovascular regeneration, which is highly relevant for conditions like neurovascular degeneration. However, despite these promising findings, the clinical application of MitoQ requires further investigation into the appropriate doses and target populations [81,82].

MitoQ is currently being assessed in a clinical trial (clinicaltrial.gov ID: NCT06930638) investigating its effects on vascular health in ischemic conditions. Preclinical results show that MitoQ reduces mitochondrial oxidative stress, improves endothelial function, elevates PGC-1α expression, and enhances angiogenesis in ischemia–reperfusion models [83] (Figure 5a).

***Compounds targeting Mitochondrial Complex III*** A cornerstone of anti-angiogenic strategies involves directly interfering with mitochondrial Complex III and curbing the generation of mitochondrial reactive oxygen species (mtROS). Disrupting Complex III effectively bottlenecks the electron transport chain, thereby altering mitochondrial membrane potential and significantly diminishing mtROS production. This disruption, in turn, cascades into downstream signaling pathways, critically impairing HIF-1α stabilization and subsequently suppressing VEGF expression and the ensuing angiogenic cascade [8]. Several compounds targeting mitochondrial Complex III and mitochondrial reactive oxygen species (mtROS) generation have demonstrated anti-angiogenic properties:

**Stigmatellin and matairesinol** reduce mtROS and block HIF-1α activation but may broadly impair electron flow, raising mitochondrial toxicity concerns. Additional studies confirm that Complex III suppression, either pharmacologically or via siRNA, impairs VEGF signaling and endothelial tube formation [3].

**UQCRB,** mitochondrial ubiquinol-cytochrome c reductase binding protein (UQCRB), a subunit of Complex III, has emerged as a key regulator of mitochondrial reactive oxygen species (mtROS)-driven angiogenesis. Its activation leads to HIF-1α stabilization and VEGF expression, promoting neovascularization both in vitro and in vivo [84]. Conversely, targeting UQCRB inhibition, rather than overexpression, has shown therapeutic promise. For instance, Terpestacin, a UQCRB-binding compound, selectively suppresses hypoxia-induced mtROS without impairing respiration, thereby attenuating HIF-1α activity and angiogenesis in multiple models [85] (Figure 5b). Importantly, downregulating UQCRB also impairs cancer stem cell-like traits in glioblastoma by reducing c-Met signaling, HIF-1α activation, and mitochondrial membrane potential, highlighting its potential in combating tumor recurrence and angiogenic niches [86]. The functional relevance of UQCRB extends beyond angiogenesis: in colon carcinoma, UQCRB-derived mtROS promote cell survival via autophagy [87], suggesting broader implications for tumor maintenance. While clinical UQCRB inhibitor data are lacking, its role as a prognostic biomarker in colorectal cancer supports further investigation [88]. Additionally, mtROS from the UQCRB site contribute to hypoxia-induced EMT and invasiveness in breast cancer, linking mitochondrial regulation to both angiogenesis and metastasis [89,90]. Transcriptional control by upstream factors like HOXD10 may further fine-tune UQCRB expression, providing another axis for therapeutic intervention [91]. Mitochondria-targeted antioxidants have seen more success. Designed to scavenge reactive oxygen species (ROS) within mitochondria, they protect against oxidative stress and maintain mitochondrial function [92]. A notable achievement is the approval of SkQ1 (Visomitin) in Russia for dry eye syndrome, demonstrating the tangible clinical benefit of this class of agents [93]. In contrast clinical trials for many mitochondrial metabolism-targeting anticancer drugs have faced significant hurdles, including dose-limiting toxicities and a narrow therapeutic index, leading to inconsistent efficacy [94] (Table 4).

### 6.2. Angiogenesis Inhibitors

**Celastrol** is a natural triterpenoid that has been shown to inhibit tumor angiogenesis by suppressing mitochondrial function and morphology through the STAT3/OPA1/P65 pathway. Opa1 (Optic Atrophy 1) is a pivotal protein orchestrating mitochondrial inner membrane fusion and the intricate organization of cristae. By specifically targeting Opa1-mediated mitochondrial fusion, celastrol effectively compromises mitochondrial integrity, directly contributing to its observed anti-angiogenic efficacy. This mechanistic elucidation provides a far more granular understanding of celastrol’s potent therapeutic actions [95].

However, celastrol pleiotropic effects (e.g., NF-κB, proteasome inhibition, [95]) make it difficult to attribute its anti-angiogenic effects solely to mitochondrial disruption. Off-target toxicity remains a challenge, necessitating mechanistic dissection and targeted delivery approaches.

**mTOR inhibitor Rapamycin** has been reported to inhibit VEGF-induced angiogenesis by interfering with mitochondrial bioenergetics and signaling. While the comprehensive impact of mTOR signaling spans far beyond angiogenesis, its central role in regulating cellular metabolism, growth, proliferation, and survival is directly pertinent to the angiogenic process. Rapamycin’s inhibition of mTOR indirectly influences these vital cellular functions, including mitochondrial bioenergetics, which are intimately intertwined with cellular growth and proliferation. By disrupting this critical metabolic link, rapamycin effectively diminishes the energy and fundamental building blocks essential for the rapid proliferation and directed migration of endothelial cells, thereby acting as a potent anti-angiogenic agent [96].

As an FDA-approved drug, rapamycin’s integration of mitochondrial and growth signaling is attractive. However, its systemic metabolic effects, coupled with feedback reactivation of mTORC2/AKT, limit its long-term utility [96] (Table 5).

Emerging therapeutic targets also include structural mitochondrial components involved in protein import and organelle integrity. The translocase of the inner mitochondrial membrane 44 (TIMM44) has recently gained attention for its essential role in maintaining mitochondrial architecture and function in ECs [31]. The inhibition of TIMM44 through either gene silencing or pharmacologic blockade markedly suppresses EC proliferation, migration, and angiogenic potential, both in vitro and in vivo [31]. These findings underscore the therapeutic potential of TIMM44 inhibition in conditions characterized by pathological angiogenesis.

## 7. Conclusions and Future Directions

Mitochondria are one of the players in the regulation of angiogenesis, with functions extending far beyond ATP production. Their roles in reactive oxygen species (ROS) signaling, metabolic flux, calcium homeostasis, and mitochondrial dynamics (including fusion, fission, and mitophagy) are essential for orchestrating endothelial cell (EC) behavior and vascular development. The mitochondrial ROS–HIF-1α–VEGF axis remains a pivotal signaling conduit linking mitochondrial activity to pro-angiogenic transcriptional programs. Moreover, emerging mitochondrial effectors such as POLRMT and TIMM44 have been identified as actionable regulators of angiogenic competence, offering promising new targets for pharmacological intervention.

While this expanding landscape of mitochondrial biology has unveiled numerous therapeutic opportunities, ranging from OXPHOS inhibition in cancer to mitochondrial enhancement in ischemic disease, several critical knowledge gaps remain:How does mitochondrial heterogeneity across different vascular beds affect angiogenic responses?

Mitochondrial density, metabolic programming, and mtDNA integrity can vary significantly between arterial, venous, capillary, and lymphatic ECs. Understanding how these differences influence responsiveness to pro- or anti-angiogenic cues remains an open question with major implications for tissue-specific therapies.

To what extent do mtDNA mutations exert context-dependent effects on angiogenesis?

While mtDNA mutations are often associated with impaired oxidative phosphorylation, some tumor models repurpose these mutations to sustain angiogenesis. Deciphering the dualistic role of mtDNA alterations in different disease environments is essential for targeting them therapeutically.

Can we selectively modulate mitochondrial dynamics (fusion, fission, mitophagy) to enhance therapeutic outcomes?

The impact of disrupting mitochondrial morphology or quality control pathways on angiogenesis is poorly defined. Whether targeting proteins like MFN2, DRP1, or PINK1 can be leveraged to fine-tune angiogenic responses is an area ripe for exploration.

How do mitochondria communicate with the nucleus and other organelles to coordinate angiogenesis?

The mitochondrial–nuclear crosstalk, including retrograde signaling and epigenetic regulation via metabolic intermediates (e.g., acetyl-CoA, succinate), remains under-characterized in endothelial contexts.

What delivery strategies can ensure safe and tissue-specific mitochondrial targeting in clinical applications?

Current approaches, such as TPP-tagged compounds or extracellular vesicle-mediated delivery, face limitations in precision, toxicity, and scalability. Developing safer and more selective delivery systems remains a priority for translational advancement.

What role does mitochondrial signaling play in the immune regulation of angiogenesis?

Recent studies suggest that mitochondrial metabolism influences not only ECs but also immune cells within the angiogenic niche. Investigating how mitochondria shape the immuno-angiogenic landscape, particularly through macrophage and neutrophil polarization, could reveal new combination strategies.

In conclusion, mitochondria represent a multifaceted and underexploited frontier in angiogenesis research. A deeper understanding of their structural diversity, metabolic plasticity, and inter organelle signaling will be key to developing next-generation therapies that can selectively promote or inhibit blood vessel formation based on specific pathological needs. Addressing the outlined questions through interdisciplinary approaches combining vascular biology, mitochondrial genomics, metabolomics, and drug delivery science will pave the way for more precise, effective, and personalized interventions in vascular medicine and oncology.

## Figures and Tables

**Figure 1 ijms-26-07960-f001:**
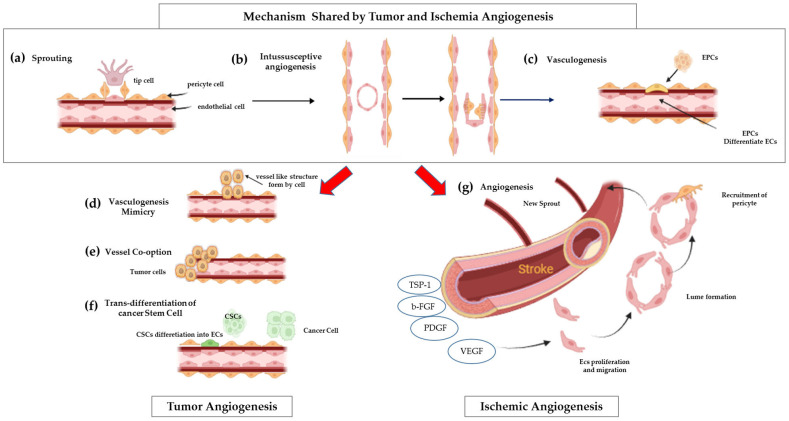
**Mechanisms of ischemia and tumor blood vessel formation**. **Shared mechanisms:** (**a**) Sprouting angiogenesis—new vessels from existing ones; (**b**) Intussusceptive angiogenesis—vessel lumen splits into two; (**c**) Vasculogenesis—bone marrow-derived EPCs differentiate into ECs to form vessels. **Tumor-specific mechanisms:** (**d**) Vascular mimicry—tumor cells form vessel-like structures; (**e**) Vessel co-option—tumors hijack existing vessels; (**f**) CSC trans-differentiation—cancer stem cells become ECs to support vasculature. As a result, tumor blood vessels are typically chaotic, twisted, and leaky. In contrast (**g**) ischemic angiogenesis produces new blood vessels well-organized, stable, and functionally mature, designed to restore proper blood flow to the affected tissue.

**Figure 2 ijms-26-07960-f002:**
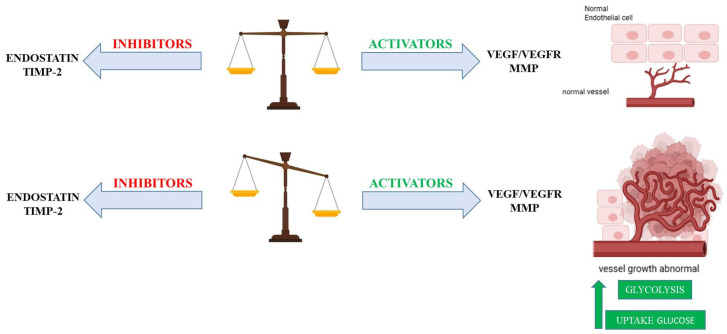
**The Angiogenic Switch in Pathological Conditions.** In pathological settings such as cancer, a tightly regulated balance between pro- and anti-angiogenic factors is disrupted, leading to the activation of the “angiogenic switch.” This process is characterized by metabolic reprogramming in endothelial cells, including increased glucose uptake and enhanced glycolysis, which fuel rapid proliferation and support the formation of new vasculatures. Tumors exploit this switch to promote neovascularization and sustain growth. Key anti-angiogenic factors include the tissue inhibitor of metalloproteinases-2 (TIMP2) and endostatin, whereas pro-angiogenic signals are driven by matrix metalloproteinases (MMPs) and the vascular endothelial growth factor and its receptor (VEGF/VEGFR).

**Figure 3 ijms-26-07960-f003:**
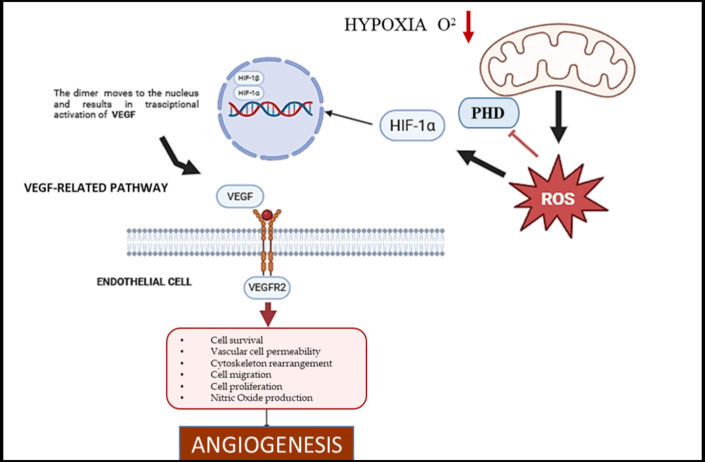
**The mtROS–HIF-1α–VEGF Axis in Hypoxia-Induced Angiogenesis.** Under hypoxic conditions, mitochondrial Complex III generates mitochondrial reactive oxygen species (mtROS), which diffuse into the cytosol and inhibit prolyl hydroxylase domain (PHD) enzymes. This inhibition stabilizes hypoxia-inducible factor 1-alpha (HIF-1α), preventing its degradation. Stabilized HIF-1α translocates to the nucleus, promoting transcription of the vascular endothelial growth factor (VEGF). The VEGF then binds to its receptor, VEGFR2, on both mature endothelial cells and circulating endothelial progenitors (CEPs), activating downstream signaling pathways such as ERK and Akt. This leads to endothelial cell proliferation, migration, and mobilization, key steps in angiogenesis.

**Figure 4 ijms-26-07960-f004:**
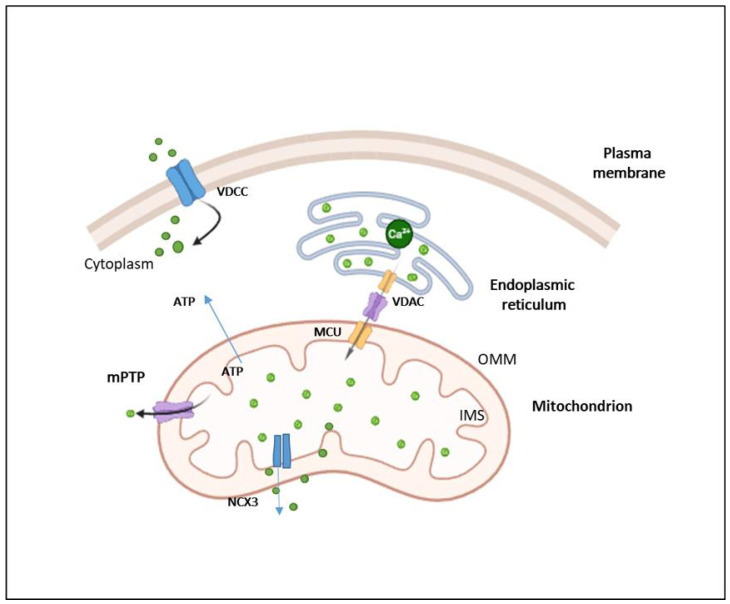
**Mitochondrial Calcium Transport and Microdomain Signaling**. Calcium transport into mitochondria begins at the outer mitochondrial membrane (OMM), where voltage-dependent anion channels (VDACs) mediate the flux of calcium ions from the cytosol into the intermembrane space (IMS). Calcium then traverses the inner mitochondrial membrane (IMM) via the mitochondrial calcium uniporter complex (MCU), reaching the mitochondrial matrix. Calcium efflux mechanisms include the mitochondrial Na^+^/Ca^2+^ exchanger (NCX3) and the mitochondrial permeability transition pore (mPTP). Mitochondria form specialized high calcium microdomains at contact sites with the endoplasmic reticulum (ER) and the plasma membrane (PM). These regions exhibit elevated calcium concentrations due to localized calcium entry, often facilitated by voltage-dependent calcium channels (VDCCs) at the PM, enabling efficient calcium signaling and homeostasis within the cell.

**Figure 5 ijms-26-07960-f005:**
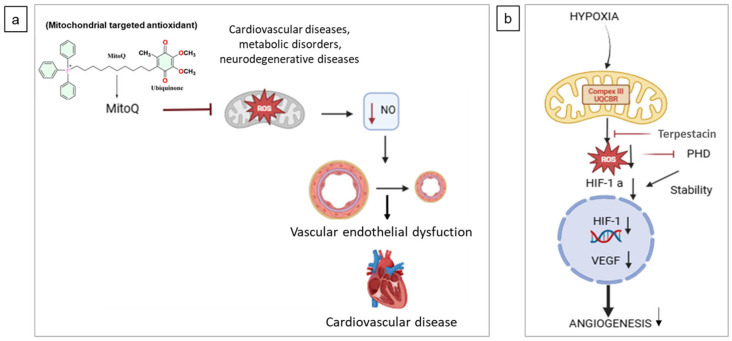
**MitoQ and UQCRB antioxidant action.** (**a**) MitoQ, a mitochondria-targeted CoQ10 derivative, acts as a potent antioxidant that preserves mitochondrial function and reduces oxidative stress. By limiting ROS production, it may help maintain nitric oxide signaling, prevent endothelial dysfunction, and lower cardiovascular disease (CVD) risk, which is driven by oxidative stress, inflammation, and vascular abnormalities; (**b**) UQCRB acts as a mitochondrial oxygen sensor in Complex III. Terpestacin inhibits hypoxic signaling by targeting UQCRB, reducing ROS production, HIF-1α stabilization, and VEGF expression, thereby suppressing tumor angiogenesis.

**Table 2 ijms-26-07960-t002:** Mitochondrial Dynamics and quality control in Angiogenesis.

**Drp1/Fis1 (fission)**	Generates mitochondrial fragments, which increase mtROS, inducing HIF-1α/VEGF activation and promoting sprouting angiogenesis. In tumor cells, hyperactive Drp1 enhances its own angiogenesis and remodels the microenvironment.	[34,41,44]
**Opa1/Mfn1/2 (fusion)**	Promotes network elongation, which decreases mtROS, dampening angiogenic signals.	[34,41,45,47]
**Mitochondrial trafficking**	Relocates mitochondria to EC leading edge, supporting migration via ATP/ROS supply.	[44]
**Mitophagy & quality control**	PINK1/FUNDC1/BINP3 pathways regulate vessel density and sprouting through organelle turnover.	[48]

**Table 3 ijms-26-07960-t003:** Reconciling Pro-angiogenic and Anti-angiogenic roles of mtDNA Mutations.

	mtDNA Mutational Effect	Angiogenic Outcome	Ref
**Pulmonary vascular disease/PAH**	NFU1 mutation induces Fe-S cluster dysfunction causing ROS imbalance	Impaired angiogenesis reduces capillary density; rescued by metabolic cofactors	[61]
**Cardiovascular disease** **(e.g., CAD variant)**	tRNA^Thr^ mutation induces mitochondrial instability	Impaired EC survival and angiogenesis	[57]
**Tumor context, high-heteroplasmy**	Disrupted respiratory gene variants cause glycolytic switch	Enhanced sensitivity to anti-angiogenic therapy; possibly reduced angiogenesis	[66,67]
**Tumor context, low-heteroplasmy or rescue**	mtDNA transfer or specific pathogenic mutations	Increased VEGF/MMP/CXCL12 expression, angiogenesis, and metastasis	[65,67]

**Table 4 ijms-26-07960-t004:** Therapeutic Benefits vs. Potential Risks in the contest of both ischemic disease and cancer.

Context	Target/Agent	Benefit	Limitations/Risks	Ref.
Ischemic Disease	MitoQ	Reduces mtROS Enhances endothelial recovery	Dosing and long-term safety TBD	[80,81]
Cancer	UQCRB inhibitors	Blocks VEGFR2/HIF-1α Reduces CSC traits and EMT signals	No clinical trials yet; potential systemic toxicity unknown	[84,89]
	Complex III inhibitors	Suppress tumor angiogenesis via mtROS reduction	May impair normal mitochondrial function; off-target effects	[8]

**Key Concern: Trade-offs;** inhibiting mitochondrial activity may benefit cancer therapy but inadvertently impair angiogenic repair in ischemic tissues.

**Table 5 ijms-26-07960-t005:** Comparative overview of antiangiogenesis compounds.

Agent	Target	Mechanism	Strengths	Limitations/Challenges	Ref
**Terpestacin**		Inhibits HIF-1α/VEGF activation by reduction of hypoxia-driven mtROS production	Targeted mtROS inhibition without disrupting respiration	Needs validation in chronic diseases; EC-specific effects unknown	[84,85]
**Stigmatellin/** **Matairesinol**	Complex III Qo/Qi	Broad inhibition of electron flow that reduces mtROS production	mtROS suppression confirmed	High mitochondrial toxicity; limited in vivo efficacy data	[3]
**Celastrol**	Mito structure + AKT/mTOR	Mitochondrial disruption; AKT/mTOR inhibition	Multi-pathway anti-angiogenic and antitumor activity	Non-specific effects; limited mitochondrial-specific data	[95]
**Rapamycin**	mTORC1	Impairs mitochondria via mitophagy suppression	Has clinical approval; effective in endothelial models	Broad metabolic impact; compensatory	[96]

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
