# Peer review of "The Multifaceted Role of Mitochondria in Angiogenesis"

_ijms, 2025, doi:10.3390/ijms26167960_

Round 1

Reviewer 1 Report

Comments and Suggestions for Authors

This review article provides a comprehensive and insightful exploration of the diverse roles mitochondria play in angiogenesis, extending far beyond their traditional function as cellular powerhouses. The authors effectively synthesize a vast body of research to highlight the intricate mechanisms by which mitochondrial metabolism, signaling, and dynamics influence endothelial cell behavior and vascular formation.Below are a few specific comments that should be addressed to further improve the quality of the manuscript: 

1.Please ensure consistency in terminology, such as avoiding the mixing of "mtROS" and "mROS". 
2.Please correct spelling errors, such as the incorrect spelling of "Mithocondria" in the title and keywords, which should be "Mitochondria".
3.The logical framework of the "bidirectional effect" of mtDNA mutations on angiogenesis can be further clarified.
4.The presentation and critical interpretation of results of previous studies should be improved, integrating these findings in the narrative of the present review.

Author Response

The authors would like first to thank the Reviewers for the time taken to review the manuscript and most of all for his important comments and recommendations, that helped to significantly improve the manuscript.

The authors have followed the Reviewer recommendations and changed the manuscript accordingly.

A point by point reply to the Reviewer’s comments is provided below, within the original text:

Reviewer 1

Reviewer: This review article provides a comprehensive and insightful exploration of the diverse roles mitochondria play in angiogenesis, extending far beyond their traditional function as cellular powerhouses. The authors effectively synthesize a vast body of research to highlight the intricate mechanisms by which mitochondrial metabolism, signaling, and dynamics influence endothelial cell behavior and vascular formation.

Authors: We would like to thank the Reviewer for his kind positive comment about the relevance of the manuscript.

Specific comments which have been addressed as requested:

  1. Please ensure consistency in terminology, such as avoiding the mixing of "mtROS" and "mROS".

Authors: Terminology has been consistently modified as requested

  1. Please correct spelling errors, such as the incorrect spelling of "Mithocondria" in the title and keywords, which should be "Mitochondria".

Authors: We apologize for the inconvenient. Spelling has been corrected.

3.The logical framework of the "bidirectional effect" of mtDNA mutations on angiogenesis can be further clarified.

Authors:  We thank the reviewer for his/her important request that helped us to further clarify this key issue.  It has been now clarified in the text how the consequences of mtDNA mutations on angiogenesis vary based on the mutation class, tissue type, heteroplasmy level, and availability of compensatory metabolic programs. The discussion has been focused on two different environment ischemia angiogenesis versus tumor angiogenesis. (Please see the revised section 5. The Impact of Mitochondrial DNA mutation on Angiogenesis: When Power Fails) and in the new table 3.

4.The presentation and critical interpretation of results of previous studies should be improved, integrating these findings in the narrative of the present review.

Authors:  Thanks for the request that helped to further improve the manuscript. We have integrated the presentation and critical interpretation of results from previous studies in sections 4, 5 and 6 of the review.

In conclusion, the authors would like to thank the Reviewers for their dedication in reading and commenting our manuscript, and for their comments and recommendations which we believe have greatly helped to improve the manuscript. We hope that its revised version could be considered satisfactory for publication.

Sincerely – The corresponding author

Reviewer 2 Report

Comments and Suggestions for Authors

After review of the manuscript entitled "The Multifaceted Role of Mitochondria in Angiogenesis" I may suggest the following comments to be addressed by the Authors:

1) The review has a certain degree of novelty as is provides a fine overview of recent proceedings in the field yet the crucial claim of the authors "Mitochondria serve as central regulators of angiogenesis, contributing far beyond their canonical role in ATP production" is a controversial point. Current understanding of vascular physiology is based on several concepts including tissue specificity (including tumor vasculature), complex paracrine and transcriptional regulation and cross-talk of endothelial cells with numerous other elements of the tissue. This basically undermines that a "pivotal" or "central" element exists (except for endothelial cells which are indespensable in vasculature). Thus, claims of such magnitude are unlikely to be well-met by the Reader

2) Authors clearly stipulate the narrative nature of the work yet it is more of chapter in a textbook style covering basic features of each aspect. Overall, this provides a quality read for a newcome specialist in vascular biology and angiogenesis while a series of reviews on the topic has been published with significant detail to the point.

  • DOI: 10.1007/s11033-018-4488-x
  • 10.1007/978-3-319-55330-6_21
  • 10.1186/s12967-023-04286-1

    Thus, while giving a good overview hardly we can ensure that is expands the knowledge in the field

3) The review lacks illustrations to make a clearer overview of aspects covered in the text. Certain images are good for a lecture yet lack addressing and connection with details provided in review's subsections.

Overall, the review is subject to being revised and improved so I'd be happy to re-review it once required.

Regards, Reviewer

Author Response

The authors would like first to thank the Reviewers for the time taken to review the manuscript and most of all for his important comments and recommendations, that helped to significantly improve the manuscript.

The authors have followed the Reviewer recommendations and changed the manuscript accordingly. A point by point reply to the Reviewer’s comments is provided below, within the original text:

Reviewer 2

Reviewer: The review has a certain degree of novelty as is providing a fine overview of recent proceedings in the field yet the crucial claim of the authors "Mitochondria serve as central regulators of angiogenesis, contributing far beyond their canonical role in ATP production" is a controversial point. Current understanding of vascular physiology is based on several concepts including tissue specificity (including tumor vasculature), complex paracrine and transcriptional regulation and cross-talk of endothelial cells with numerous other elements of the tissue. This basically undermines that a "pivotal" or "central" element exists (except for endothelial cells which are indispensable in vasculature). Thus, claims of such magnitude are unlikely to be well-met by the Reader

Authors: We agree with the reviewer and we had changed the sentence "Mitochondria serve as central regulators of angiogenesis, contributing far beyond their canonical role in ATP production" as follows: “Mitochondria are one of the players in the regulation of angiogenesis, with functions extending far beyond ATP production.” (See first paragraph of the conclusion section)

Reviewer: Authors clearly stipulate the narrative nature of the work yet it is more of chapter in a

textbook style covering basic features of each aspect. Overall, this provides a quality read for a

newcome specialist in vascular biology and angiogenesis while a series of reviews on the topic has

been published with significant detail to the point.

 DOI: 10.1007/s11033-018-4488-x

 10.1007/978-3-319-55330-6_21

 10.1186/s12967-023-04286-1

Authors: We thank the Reviewer for stating that the present review “provides a quality read for a newcome

specialist in vascular biology and angiogenesis”.

We agree with the reviewer that previous reviews on the topic has been also published with significant

detail to the point. To this regard, we would like to confirm that the following previous “reviews” indicated

by the reviewer have been incorporated in the references as suggested (see references 3, 4 and 5) :

  • DOI: 10.1007/s11033-018-4488-x
  1. a) Reichard A, Asosingh K. The role of mitochondria in angiogenesis. Mol Biol Rep. 2019

Feb;46(1):1393-1400.

  • 10.1007/978-3-319-55330-6_21
  1. b) Marcu R, Zheng Y, Hawkins BJ. Mitochondria and Angiogenesis. Adv Exp Med Biol. 2017;982:371-

406.

  • 10.1186/s12967-023-04286-1
  1. c) Luo Z, Yao J, Wang Z, Xu J. Mitochondria in endothelial cells angiogenesis and function: current

understanding and future perspectives. J Transl Med. 2023 Jul 5;21(1):441.

Reviewer: Thus, while giving a good overview hardly we can ensure that is expands the knowledge in the

field

Authors: We thank the Reviewer for considering our manuscript “a good overview”. We also agree with the

reviewer that being the nature of the manuscript a review it will hardly expands the knowledge in the field,

instead providing a quality read for a newcome specialist in vascular biology and angiogenesis, as the

reviewer has kindly stated in his/her initial statement of this point.

Reviewer:  The review lacks illustrations to make a clearer overview of aspects covered in the text. Certain images are good for a lecture yet lack addressing and connection with details provided in review's subsections.

Authors: We fully agree with the reviewer that manuscript could benefit of additional illustrations to better clarify the content of the review which are now included in the manuscript (new Figure 1 and 5, new tables 2,3,4, and 5 )

In conclusion, the authors would like to thank the Reviewers for their dedication in reading and commenting our manuscript, and for their comments and recommendations which we believe have greatly helped to improve the manuscript. We hope that its revised version could be considered satisfactory for publication.

Sincerely – The corresponding author

Reviewer 3 Report

Comments and Suggestions for Authors

This review offers a clear and well-organized overview of the multifaceted roles of mitochondria in angiogenesis, highlighting their contributions beyond metabolism to signaling, redox regulation, calcium homeostasis, and quality control. It effectively links mitochondrial biology to endothelial cell function and angiogenesis, making it a valuable resource for researchers in vascular biology, cancer, and metabolic diseases. The writing is clear and scholarly, with appropriate citations to support key arguments. However, some sections could benefit from greater mechanistic depth, updated references, and a more critical discussion of conflicting findings in the field.

Summary of Issues Identified in the Manuscript Review

  1. Typographical Error: The term “Mithocondrion” should be corrected to “mitochondrion” (singular) or “mitochondria” (plural) as appropriate. “Mithocondrial Calcium Trasport” should also be corrected to “Mitochondrial Calcium Transport”.
  2. Calcium Channel Terminology: In Figure 3, the terms “voltage-dependent calcium channel (VDCC)” and “voltage-operated calcium channel (VOCC)” are used interchangeably. According to IUPHAR and NCBI databases, the standardized term is “VDCC.” The term “VOCC” should be replaced with “VDCC” for consistency.
  3. Mechanistic Integration: While the metabolic interplay (glycolysis vs. OXPHOS) is well-covered, the discussion on mitochondrial dynamics (fusion/fission) and their direct impact on angiogenesis remains relatively superficial. A deeper exploration of how these dynamics influence angiogenesis would be beneficial.
  4. Lack of Critical Discussion on Controversial Findings: The manuscript presents mtDNA mutations as both pro- and anti-angiogenic but does not sufficiently reconcile these opposing observations. A deeper analysis of tumor-specific vs. ischemic angiogenesis contexts would strengthen this section.
  5. The impact of mitochondrial transfer via tunneling nanotubes and extracellular vesicles on angiogenesis is an emerging and potentially significant topic that is not addressed in the manuscript. Including this could provide a more comprehensive view of mitochondrial contributions to angiogenesis.
  6. The conclusion section could be more forward-looking by proposing specific unanswered questions, such as how mitochondrial heterogeneity across different vascular beds affects angiogenesis.
  7. Enhance the therapeutic section: The therapeutic section could be enhanced by including information on ongoing clinical trials, such as the use of MitoQ in ischemic conditions and UQCRB inhibitors in cancer, to provide a more up-to-date and comprehensive overview of the field.
  8. Inconsistencies in Reference Formatting

1) Inconsistent application of “et al.” for multi-author references. For example, Reference 6 (Xie et al.) uses “et al.,” while other references list all authors.

2) Missing Page Numbers: References 42, 44, 48, and 51 lack page numbers.

3) Inconsistent Journal and Volume Formatting

For example, Reference 18 is formatted as “2012, 3:412,” whereas Reference 20 appears as “Int. J. Mol. Sci. 2023, 24, 2086.” The references should adhere to a consistent format throughout the manuscript.

4) Inconsistent Hyphen Usage for Page Ranges:  Some references use a short hyphen “-” (e.g., Reference 24), while others use an en dash “–” (e.g., Reference 27). It is recommended to standardize the use of the short hyphen “-” for page ranges.

5) Content-Related issue: Discrepancy in Reference Content: In lines 196-198, the text mentions “SIRT3,” but Reference 23 cites “STAT3.” This discrepancy needs to be resolved.

Author Response

The authors would like first to thank the Reviewers for the time taken to review the manuscript and most of all for his important comments and recommendations, that helped to significantly improve the manuscript.

The authors have followed the Reviewer recommendations and changed the manuscript accordingly.

A point by point reply to the Reviewer’s comments is provided below, within the original text:

Reviewer 3

Reviewer: This review offers a clear and well-organized overview of the multifaceted roles of mitochondria in angiogenesis, highlighting their contributions beyond metabolism to signaling, redox regulation, calcium homeostasis, and quality control. It effectively links mitochondrial biology to endothelial cell function and angiogenesis, making it a valuable resource for researchers in vascular biology, cancer, and metabolic diseases. The writing is clear and scholarly, with appropriate citations to support key arguments

Authors: We would like to thank the Reviewer for his/her kind positive comment about the relevance of the manuscript. In particular, for stating that it is a valuable resource for researchers in vascular biology, cancer, and metabolic diseases.

Reviewer: However, some sections could benefit from greater mechanistic depth, updated references, and a more critical discussion of conflicting findings in the field.

Authors: Thanks for the request that helped to further improve the manuscript. We have followed the reviewer’s suggestions in the revised section 4 and 5 of the manuscript

Specific comments which have been addressed as requested:

  1. Typographical Error: The term “Mithocondrion” should be corrected to “mitochondrion” (singular) or “mitochondria” (plural) as appropriate. “Mithocondrial Calcium Trasport” should also be corrected to “Mitochondrial Calcium Transport”.

Authors: We apologize for the inconvenient. Typographical Errors have been corrected.

  1. Calcium Channel Terminology: In Figure 3, the terms “voltage-dependent calcium channel (VDCC)” and “voltage-operated calcium channel (VOCC)” are used interchangeably. According to IUPHAR and NCBI databases, the standardized term is “VDCC.” The term “VOCC” should be replaced with “VDCC” for consistency.

Authors: The term “VOCC” has been replaced with “VDCC”

  1. Mechanistic Integration: While the metabolic interplay (glycolysis vs. OXPHOS) is well-covered, the discussion on mitochondrial dynamics (fusion/fission) and their direct impact on angiogenesis remains relatively superficial. A deeper exploration of how these dynamics influence angiogenesis would be beneficial.

Authors: Thanks for the request that helped to further improve the manuscript. The effects of mitochondrial dynamics on angiogenesis are now explored and added to the revised section 4 of the manuscript (sub section 4.3 and new table 2).

  1. Lack of Critical Discussion on Controversial Findings: The manuscript presents mtDNA mutations as both pro- and anti-angiogenic but does not sufficiently reconcile these opposing observations. A deeper analysis of tumor-specific vs. ischemic angiogenesis contexts would strengthen this section.

Authors:  We thank the reviewer for his/her important request that helped us to further clarify this key issue.  It has been now clarified in the text how the consequences of mtDNA mutations on angiogenesis vary based on the mutation class, tissue type, heteroplasmy level, and availability of compensatory metabolic programs. The discussion has been focused on two different environments: ischemia angiogenesis versus tumor angiogenesis. (Please see the revised section 5. The Impact of Mitochondrial DNA mutation on Angiogenesis: When Power Fails and the new table 3)

  1. The impact of mitochondrial transfer via tunneling nanotubes and extracellular vesicles on angiogenesis is an emerging and potentially significant topic that is not addressed in the manuscript. Including this could provide a more comprehensive view of mitochondrial contributions to angiogenesis.

Authors:  Thank you very much for emphasising the importance of including the impact of extracellular vescicles and mitochondrial transfer via tunneling nanotubes on angiogenesis. Such impact is now analized in the new subsections 4.5 and 5.4 of the manuscript.

6.The conclusion section could be more forward-looking by proposing specific unanswered questions, such as how mitochondrial heterogeneity across different vascular beds affects angiogenesis.

Authors: The conclusion has been changed as suggested.

  1. Enhance the therapeutic section: The therapeutic section could be enhanced by including information on ongoing clinical trials, such as the use of MitoQ in ischemic conditions and UQCRB inhibitors in cancer, to provide a more up-to-date and comprehensive overview of the field.

Authors:  The information on ongoing clinical trials, such of MitoQ in ischemic conditions and UQCRB inhibitors in cancer have now been discussed in the revised section 6 and in the new tables 4 and 5.

  1. Inconsistencies in Reference Formatting

Authors. We have reformatted the references indicated as kindly requested.

Furthermore, we have changed the former reference 23 with the new reference 24 He, X.; Zeng, H.; Chen, J. X., Emerging role of SIRT3 in endothelial metabolism, angiogenesis, and cardiovascular disease. J Cell Physiol 2018, 234, (3), 2252-2265.    

In conclusion, the authors would like to thank the Reviewers for their dedication in reading and commenting our manuscript, and for their comments and recommendations which we believe have greatly helped to improve the manuscript. We hope that its revised version could be considered satisfactory for publication.

Sincerely – The corresponding author

Round 2

Reviewer 1 Report

Comments and Suggestions for Authors

The author has made revisions point by point, and the manuscript has met the standard for acceptance.

Author Response

The authors would like to thank the reviewers once again for the time dedicated for reviewing the manuscript and, most of all, for their important comments and suggestions, which have significantly contributed to its improvement.

Sincerely 

The corresponding author

Reviewer 2 Report

Comments and Suggestions for Authors

After revision I have no further queries on the manusciprt

Author Response

(The authors gave the same response as above.)

Reviewer 3 Report

Comments and Suggestions for Authors

Thank you for carefully revising the manuscript. Upon checking the latest files, I noticed two minor items that still need your attention before the manuscript can be finalized.

  1. Missing page numbers in references. For example,references 1,12,21,30. Please provide the full page range and check the others.
  2. In Figure 4, the word “mitochondria” is misspelled.
  3. One quick question: Table 1 contains references; should the same practice be applied to the remaining tables?

Author Response

The authors would like to thank again the Reviewers for the time taken to review the manuscript and most of all for his important comments and recommendations, that helped to significantly improve the manuscript.

The authors have followed the Reviewer recommendations and changed the manuscript accordingly.

A point-by-point reply to the Reviewer’s comments is provided below, within the original text:

Reviewer 3

Specific comments which have been addressed as requested:

  1. Missing page numbers in references. For example, references 1,12,21,30. Please provide the full-page range and check the others.

Authors:  We apologize for the inconvenient. We have corrected the references indicated as kindly requested and we have checked the others.

All the corrected references are highlighted in yellow.

  1. In Figure 4, the word “mitochondria” is misspelled.

Authors: Spelling has been corrected.

  1. One quick question: Table 1 contains references; should the same practice be applied to the remaining tables?

Authors:  We agree with the reviewer and therefore we have integrated tables 2,3,4 and 5 with the appropriated references.

In conclusion, the authors would like to thank the Reviewers for their dedication in reading and commenting our manuscript, and for their comments and recommendations which we believe have greatly helped to improve the manuscript.

Sincerely  

The corresponding author